# COVID-19 Vaccine Booster Hesitancy among Healthcare Workers: A Retrospective Observational Study in Singapore

**DOI:** 10.3390/vaccines10030464

**Published:** 2022-03-17

**Authors:** Sky Wei Chee Koh, Hwei Ming Tan, Wayne Han Lee, Jancy Mathews, Doris Young

**Affiliations:** 1National University Polyclinics, National University Health System, Singapore 609606, Singapore; hwei_ming_tan@nuhs.edu.sg (H.M.T.); wayne_han_lee@nuhs.edu.sg (W.H.L.); jancy_mathews@nuhs.edu.sg (J.M.); 2Division of Family Medicine, Yong Loo Lin School of Medicine, National University of Singapore, Singapore 119228, Singapore; mdcydyl@nus.edu.sg

**Keywords:** COVID-19 vaccine, vaccine hesitancy, healthcare workers, primary care, general practice, Singapore

## Abstract

Background: COVID-19 booster uptake remained poor among healthcare workers (HCW) despite evidence of improved immunity against Delta and Omicron variants. While most studies used a questionnaire to assess hesitancy, this study aimed to identify factors affecting booster hesitancy by examining actual vaccine uptake across time. Method: COVID-19 vaccination database records among HCW working at seven Singaporean public primary care clinics between January to December 2021 were extracted, with sex, profession, place of practice, vaccination type, and dates. Time to booster was calculated from the date of vaccination minus date of eligibility. Chi-square test was used to compare the relationship between first dose and booster hesitancy, Kaplan–Meier method and log-rank test were adopted to evaluate differences in cumulative booster uptake. Multivariate Cox regression was used to investigate predictors for timely booster vaccination. Vaccination rate was charted across time and corroborated with media releases pertaining to legislative changes. Results: A total of 877 of 891 (98.9%) primary care HCW were fully vaccinated, 73.8% of eligible HCW had taken the booster. HCW were less booster hesitant [median 16 (5–31.3) days] compared to the first dose [median 39 (13–119.3) days]. First dose-hesitant HCW were more likely to be booster hesitant (OR = 3.66, 95%CI 2.61–5.14). Adjusting for sex, workplace, and time to first dose, ancillary (HR = 1.53, 95%CI 1.03–2.28), medical (HR = 1.8, 95%CI 1.18–2.74), and nursing (HR = 1.8, 95%CI 1.18–2.37) received boosters earlier compared with administrative staff. No temporal relationship was observed between booster uptake, legislative changes, and COVID-19 infection numbers. Conclusion: Vaccine hesitancy among HCW had improved from first dose to booster, with timely booster vaccination among medical and nursing staff. Tailored education, risk messaging, and strategic legislation might help to reduce delayed booster vaccination.

## 1. Introduction

As of February 2022, more than 433 million individuals worldwide have been infected with Coronavirus Disease 2019 (COVID-19), with close to 6 million deaths since its emergence in late 2019 [1]. Vaccination is a key public health strategy because it has been shown to be effective in reducing the risk of infection and severe disease [2,3]. We have witnessed how the process of vaccine development from in vivo testing to population-wide implementation occurred in just under one year, with the first doses given in December 2021. Vaccine hesitancy is defined by the World Health Organization (WHO) as the delay in acceptance or refusal of vaccination despite availability [4]. In the current COVID-19 pandemic, this has resulted in a significant global health threat with negative socioeconomic and health effects on individuals and their communities [5]. Understandably, the pre-conceived novelty, speed of development, and lack of information regarding long-term sequelae of COVID-19 vaccines, coupled with special approvals to acquire rapid access due to the current pandemic, played a role in increased hesitancy compared to previous immunizations.

Healthcare workers (HCW) are defined as paid or unpaid persons engaged in actions whose primary intent is to enhance health [6]. They are at increased risk of exposure due to the nature of their work [7] thus achieving high vaccination rates with timely booster doses in this group is critical. As the first group to be vaccinated and imbued with adequate knowledge, HCW were often looked upon as highly trusted sources of guidance about COVID-19 vaccination [8]. HCW were therefore best positioned to share locally credible experiences to be role models for the rest of the community. However, levels of vaccine hesitancy among HCW were comparable to that of the general population across different countries [9]. Hesitancy among HCW was lower among doctors, self-perceived high risk and involved with care for COVID-19 patients and improved confidence, with a greater understanding of risks and side effects across time [10,11,12,13].

An island city-state with close to 360,000 COVID-19 infections as of February 2022, Singapore managed to reduce community spread and kept its death rate low with a high vaccination rate of 91% [14]. This was achieved through legislation such as vaccine-differentiated safe management measures (VDS), which allowed fully vaccinated individuals to visit shopping malls, dine out at restaurants and food centers, and carry on with day-to-day social activities (akin to the vaccine green pass in many countries worldwide). This was also supported by evidence-based decision making showcased by routine updates to guidelines by a COVID-19 multi-ministry task force, suitable communications, and strong primary care. Among HCW, vaccine hesitancy rates were low due to high self-perceived risk, similar to many studies performed abroad [15]. Males working in healthcare, ethnicity, and age were associated with increased vaccination uptake [16].

As new variants such as Delta and Omicron emerge, booster doses have been shown to confer greater protection by improving immunity for already fully vaccinated individuals [17,18]. Because of this, some countries chose to strengthen and improve coverage of primary vaccination series, while others pushed for non-compulsory booster vaccination. In Singapore, booster doses for mRNA COVID-19 vaccines were first introduced in September 2021 and were non-compulsory (not included in the requirement for fully vaccinated status). Those aged 12 and above are now eligible to receive their mRNA booster 5 months after completion of their primary vaccination series. However, only 58% of the total population have received their booster, compared to 67% among a public healthcare cluster. It would be interesting to observe the response to booster vaccine uptake before it became a requirement to maintain fully vaccinated status for the purpose of VDS, which was announced in early January for nationwide implementation from February 2022.

Given previous COVID-19 vaccination experience, we hypothesized lesser booster hesitancy among HCW with shorter lag time compared to previous vaccination, and uptake would be influenced by vaccinated-related legislative changes. However, multiple questionnaire-based cross-sectional studies conducted worldwide found conflicting evidence pertaining to factors associated with COVID-19 vaccine hesitancy [9]. This could be attributed to individual subjectivity, social desirability and non-response bias among vaccine-hesitant individuals, difficulties in measuring hesitancy levels through concrete Likert scales, and inability to monitor across a time continuum while accounting for differences in contextual and socio-cultural influences between countries [19,20]. To overcome this conundrum, a longitudinal observational study to follow a subset of individuals through time would be helpful, and using time to vaccination from eligibility would improve accuracy in measuring hesitancy level. Hence, this study aims to identify factors affecting booster hesitancy by examining actual vaccine uptake across time and triangulating this with the timing of media announcements pertaining to legislative changes.

## 2. Methods

The study was a retrospective observational study reviewing the prevalence, trend, and factors affecting COVID-19 vaccine and booster uptake among employees in a healthcare cluster in Singapore.

### 2.1. Study Population and Sampling

COVID-19 vaccination and booster records of HCW working in a Singaporean public primary healthcare institution, comprising 891 staff from 7 primary care clinics and central office, were extracted from 1 January 2021 to 10 December 2021. The inclusion criteria for the study population were physicians from medical and dental departments, nurses, allied health professionals, and ancillary staff. The exclusion criteria were temporary staff, doctors doing clinical rotations and attachments, staff from pharmacy and diagnostics services within the primary care clinics as they were not employed by the same public primary healthcare institution. Records of COVID-19 vaccination, including brand and vaccination dates, sex, profession, and place of practice, were extracted anonymously from the Staff Surveillance System (S3), a one-stop database to track staff immunizations and surveillance used by public healthcare institutions in Singapore. It was an institutional mandate to input and update vaccination records as this determined eligibility for work in high-risk areas. This determined the deployment of HCW and had to be submitted to the Government of Singapore to ensure workplace safety, therefore ensuring its accuracy.

### 2.2. Study Setting

There were 4 COVID-19 vaccine brands (Pfizer-BioNTech/Comirnaty, Moderna, CoronaVac, Sinopharm BIBP) approved for use in Singapore. The National Vaccination Program recommended by the Expert Committee on COVID-19 Vaccination (EC-19V) recognized 2 mRNA (Pfizer-BioNTech/Comirnaty or Moderna) or 3 non-mRNA (CoronaVac, Sinopharm BIBP) vaccines as complete primary vaccination series.

HCW vaccination exercise within the institution started on 8 January 2021 with the Pfizer-BioNTech/Comirnaty vaccine, with adequate slots provided for HCW to undertake their vaccinations at their respective clinics across a 6-week duration. To ensure availability and accountability of vaccines, these slots were paired appointment dates for 2 consecutive COVID-19 vaccine doses 3–4 weeks apart (i.e., HCW who undertook COVID-19 vaccine dose 1 will have a guaranteed dose 2 appointment in 3–4 weeks).

For the purpose of this study, a booster is defined as vaccination after completion of primary vaccination series. Based on EC-19V recommendations on 10 September 2021, a booster should be received 6 months after the mRNA primary vaccination series or 3 months after the 3rd dose for the non-mRNA primary vaccination series. Invitations were sent to those all aged 60 and above or immunocompromised, and booster vaccination commenced on 15 September 2021. This was extended to all aged 50 to 59 on 4 October 2021. Subsequently, on 9 October 2021, booster vaccination was extended to all HCW in all clinics. Booster eligibility was brought forward 5 months after the 2nd COVID-19 vaccine dose (announced on 20 November 2021) due to evidence of waning antibodies. As HCW had different booster eligibility dates, COVID-19 vaccination had become easily available with the ability to walk into every clinic. There were no administrative or logistical difficulties in acquiring boosters. On 6 January 2022, it was announced that the booster dose would be required to maintain fully vaccinated status for the purpose of VDS for 270 days after the last dose of the primary vaccination series, from 14 February 2022 onward in Singapore.

### 2.3. Study Definitions and Parameters

To ensure strict confidentiality of HCW and clinic reputation and identity, the 7 clinics were anonymized to Clinics A to G. To account for vaccine supply availability during initial vaccination service rollout for the first dose of the COVID-19 vaccine. We defined COVID-19 1st dose vaccine hesitancy as the failure of uptake within 6 weeks of vaccine introduction during the HCW vaccination exercise. This started on 8 January 2021. Time to dose 1, defined by the time taken (in days) for 1st COVID-19 vaccination dose uptake from day of availability (8 January 2021), was calculated from HCW vaccination date. Time to dose 2, time taken (in days) from 2nd dose to 1st dose, was mostly fixed at 21–28 days as paired appointments were made to ensure vaccine availability. Time to booster is defined as the time taken (in days) for booster uptake from date of eligibility. This date of eligibility was defined as 6 months after 2nd COVID-19 dose (shortened to 5 months after 20 November 2021), or date of availability (15 September 2021 for HCW aged 60 and above, 4 October 2021 for HCW age 50–59, and 9 October 2021 for the rest of HCW), whichever was later. For example, a 32-year-old HCW who took primary vaccination series on 15 January and 5 February 2021, respectively, will be due for booster 6 months after dose 2: 4 August 2021. Booster would not have been possible then as it was only made available from 9 October 2021. This HCW proceeded on to take booster on 14 October 2021, making his time to booster (from eligibility) to be 5 days. Therefore, COVID-19 booster hesitancy was defined as failure to receive COVID-19 booster despite eligibility, as of 10 December 2021, the last day of data collection for this study.

### 2.4. Statistical Analysis

Statistical analyses were performed with IBM^®^ SPSS^®^ Statistics Version 28.0, R was used to calculate the time between COVID-19 doses and date of eligibility for booster dose. *p*-value of <0.05 in the two-sided test was considered as statistical significance. Descriptive statistics were performed, and numerical variables were represented as mean (±SD), median (IQR), or n (%) for categorical variables. Chi-square tests were used to evaluate COVID-19 booster hesitancy as of 10 December 2021 despite eligibility. The Kaplan–Meier method was used to estimate the cumulative coverage of the COVID-19 vaccine, and log-rank tests were used to compare the difference across subgroups. Cox regression models were used to examine the predictors of delayed COVID-19 1st dose and booster vaccination. Finally, the cumulative COVID-19 booster uptake rate was plotted against time and correlated with media announcements, vaccination changes, and COVID-19 case numbers.

### 2.5. Ethical Considerations

The study, analysis, and publication of results were approved by the NHG Domain-Specific Review Board (DSRB) on 28 December 2021. This is an ethical component body.

## 3. Results

A total of 98.5% of all 891 HCW had completed their COVID-19 primary vaccination series and were fully vaccinated (Figure 1). As of 10 December 2021, 73.8% of eligible and fully vaccinated HCW had taken the booster, while 26.2% were still hesitant, despite already being eligible for the booster vaccine. Demographics of HCW sampled and proportions in terms of sex, workplace and profession are depicted in Table 1.

Among all HCW who received their COVID-19 vaccination, the mean and median time to dose 1 (in days) were 68.8 ± 71.5, 39 (13, 119.3), respectively. The mean and median time to dose 2 were 24.2 ± 11.8, 21 (21, 24) days. The mean and median time between dose 2 and booster were 238.5 ± 35.7, 248 (219, 263) days. The mean and median time to booster (from eligibility) were 19.6 ± 16.4, 16 (5, 31.3) days.

HCW who were eligible for booster (as of 10 December 2021) (n = 756) were split into two groups: hesitant (eligible but not boostered) (n = 198) and not hesitant (vaccinated) (n = 558) toward the COVID-19 booster. There were no significant differences in booster hesitancy between sexes (*p* = 0.544), workplace (*p* = 0.134) or profession (*p* = 0.299). Among 756 HCW, initial COVID-19 first dose hesitancy was significantly associated with subsequent booster hesitancy (χ^2^ = 59.9, *p* < 0.001). Compared to HCW who received COVID-19 first dose during the HCW vaccination exercise, HCW who were first dose-hesitant were 3.66 times more likely to be booster hesitant (OR = 3.66, 95% C.I. 2.61–5.14).

The log-rank test in the Kaplan–Meier method (Table 1) showed that male sex and HCW profession were associated with shorter time to COVID-19 first dose vaccination (*p* < 0.05). Pairwise comparisons revealed that medical professionals have significantly shorter time to COVID-19 first dose vaccination compared to their administrative (*p* = 0.007), ancillary (*p* < 0.001) and nursing colleagues (*p* = 0.021) (Table 2).

Both HCW workplace and profession were significantly associated with differences in time to COVID-19 booster vaccination. Clinics B and E had significantly lower median time to booster compared to Clinic C (*p* = 0.023, 0.027), Central Office (*p* = 0.027, 0.028) and Clinic G (*p* = 0.039, 0.019). Shorter median time to COVID-19 booster was significantly noted among medical and nursing HCW, compared with their administrative (*p* = 0.013, 0.009) and allied health colleagues (*p* = 0.049, 0.047) (Table 3). This amounted to a mean of 10 and a median of 18 days delay among administrative staff compared with medical HCW in receiving the COVID-19 booster.

Our Cox regression analysis yielded a significant *p*-value of 0.008 for omnibus tests of model coefficients at each step. After controlling for time to COVID-19 dose 1 vaccination, sex, and workplace, profession was the key factor in affecting time to COVID-19 booster vaccination. Compared to administrative HCW, ancillary (HR = 1.53), medical (HR = 1.8) and nursing (HR = 1.8) staff were more likely to receive the COVID-19 booster earlier (*p* < 0.05) (Table 4).

The actual numbers of HCW COVID-19 booster vaccinations (left *Y*-axis, blue columns) were plotted against time; legislative changes and weekly COVID-19 infection growth rate (right *Y*-axis, orange line) were highlighted as shown in Figure 2. Clinics were prompt in starting boosters for HCW, with increased uptake in the 4 weeks following the 9 October 2021 announcement of HCW booster eligibility. Despite high weekly COVID-19 infection rates, booster uptake rate was low initially upon rollout in the first 4 weeks for HCW aged 60 and above, signifying no temporal relationship between the two. No significant increment of booster uptake was observed with the 20 November 2021 announcement of shortening of the eligibility period to 5 months from second dose.

## 4. Discussion

This study was one of the first in measuring vaccine hesitancy by examining the absolute number and date of vaccine uptake from the HCW vaccination database, instead of measuring hesitancy levels via questionnaires by research subjects conducted by numerous studies worldwide. This ensured the accuracy of measuring the exact delay in uptake of COVID-19 dose 1, 2, and boosters. The key results included significant reductions in both mean and median time to booster compared to the first dose (Table 1). HCW who were hesitant toward the first dose were 3.6 times more likely to be hesitant toward the booster as well. Using the Cox regression model, we were able to accurately weigh the magnitude of factors affecting booster hesitancy by defining the event as booster vaccination and comparing hazard ratios. This certainly would inform other nations who might be embarking on their path toward booster vaccination.

The significant differences in the timing of booster rollout for clinics might be explained by one clinic undergoing renovations during that time; on-site vaccination exercise was halted because of this. However, this did not impact the Cox regression model, and we were still able to draw conclusions, as depicted in Table 4.

Despite evidence and media releases on improved efficacy of Pfizer-Pfizer-Moderna over Pfizer-Pfizer-Pfizer COVID-19 booster combination [21], close to 90% of HCW opted for the Pfizer booster (Figure 1). We speculated that this was because of ease of accessibility to booster at all clinics (which only carried Pfizer), which corresponded to our previous study where we saw hesitant HCW ranking ‘Constraints’ as the lowest in the 5Cs hesitancy scale [15].

Our study showed that COVID-19 booster delay was significantly shorter compared to the initial first dose delay, possibly as confidence was boosted given numerous studies, accounts from family, and experiences from their individual vaccinations. Our study also reinforced that male HCW were less vaccine hesitant toward the first dose, a factor already well documented in existing literature [11]. In particular, we found that the differences in sex for vaccine hesitancy had become insignificant for the booster dose; this was postulated to be due to developing evidence on the safety of vaccines for pregnant and breastfeeding mothers, a concern also highlighted among female HCW [22] and discovered in many hesitancy studies among HCW worldwide [9]. This went against a survey-based study during the same time period (November 2021), where lesser booster hesitancy among male HCW was found [23], showcasing how differences in vaccine communications in addressing concerns among female HCW [22] could have played a role in influencing study results conducted in different countries.

The few studies published on booster hesitancy were in concordance with our study results. In a large U.K. study, initial first dose hesitancy increased the likelihood of booster hesitancy by five times [24]. This was noted among HCW within this study but to a lesser degree (OR = 3.66), and the difference could be explained by the population group (HCW vs. public) and setting (Singapore vs. U.K.). It was also noted that hesitant booster individuals had low confidence or trust in authorities and vaccine messaging [25], low level of self-perceived susceptibility, and high level of perceived barriers [26]. This re-emphasized the need for targeted messaging and risk counseling to hesitant individuals to improve timely boosters to those who needed them the most.

Differences in vaccine hesitancy between HCW professions had been discovered previously in numerous studies, where medical and nursing professionals were found to be less hesitant compared to their ancillary and administrative colleagues [23,25]. This paper was able to value-add by quantifying the differences in mean and median times to booster and showed significant differences after adjusting for confounders. This difference can likely be explained due to better vaccine knowledge, understanding of adverse effects, and patient-facing roles adopted by doctors and nurses during the pandemic, a phenomenon also explored in previous studies [27,28].

### 4.1. Study Strengths

Despite knowing that there were other factors not accounted for in our Cox regression, the strengths were that this study had zero dropout rate as all records were being extracted from the vaccination database. Despite a short follow-up time of 2 months, we were able to detect significant differences in time to booster between HCW groups. Future studies could focus on other factors (such as age and ethnicity, as discovered in a local study) [14] for a longer follow-up period to build a more robust Cox regression model to detect differences in other non-significant factors identified.

Our study concluded in December 2021. In the following month, in January 2022, it was announced that booster dose would be required to be considered as fully vaccinated for the purpose of VDS (Singapore’s version of the green pass) come February 2022. It would have been interesting if our study period covered this change in order to thoroughly examine the impact such legislative changes would have in influencing booster hesitancy, an important factor highlighted in an Italian study [29].

### 4.2. Study Limitations

#### 4.2.1. Selection Bias

Limitations of this study included its selection bias of one primary care cluster in Singapore, and thus would not be representative of the whole primary care landscape. Compared to only 29.2% of the population having received the booster vaccination in mid-December 2021 [14], achieving 62.6% booster vaccination among primary care HCW (558 of total 891 HCW, see Figure 1) was definitely a commendable feat. Moreover, the rest of the healthcare cluster achieved this rate 1 month later, in mid-January 2022, showing that primary care HCW might have been less hesitant. Cross-referencing with results from our previous study on the same population of HCW, we speculated that this might be due to increased self-perceived risk to COVID-19 importance of protecting oneself and their family [12,15].

#### 4.2.2. Lack of Age, Other Demographic Factors, and COVID-19 Infection Status in the Database

As COVID-19 vaccination and booster rollout were based on age cut-offs, it would be appropriate to analyze the impact of HCW age on vaccine hesitancy. In addition, there were no guidelines for COVID-19 booster vaccination for recovered COVID-19 HCW who had completed their primary vaccination series. Knowing both HCW age and COVID-19 infection status would confer greater accuracy in determining eligibility date. However, both the extraction of age and previous COVID-19 status from the vaccination database did not receive ethics approval because this might make them easily identifiable and compromise confidentiality. In addition, being infected by COVID-19 with having completed the primary vaccination series would mean that additional booster doses were not required as of February 2022. Because of this, we had to estimate the longest booster eligibility duration to be 61 days (between 10 October 2021, where booster was open to all HCW regardless of age, and 10 December 2021, the end date of our study), where in actual fact this duration could have been longer due to earlier booster eligibility due to HCW age. There were other demographic factors highlighted in multiple questionnaire-based vaccine hesitancy studies that were not accounted for in our Cox regression model, such as education level, marital status, race, socioeconomic status, and even information sources individuals trust [29,30,31,32]. Mixed studies could be performed to discover the main reasons for vaccination (such as to protect their family and friends) [33] and placed in an all-encompassing multivariate model to examine how these different factors interplay in influencing booster hesitancy.

## 5. Conclusions

Findings suggested that COVID-19 booster vs. initial first dose hesitancy among HCW had improved significantly, with close to 75% of HCW receiving booster in just 2 months upon its release. This would presumably be due to ease of accessibility, improved confidence with previous own experience of having received the vaccine, and evidence and guidelines on safety profile. This study validated previous questionnaire studies on factors affecting vaccine hesitancy, showcasing significant delay in time to COVID-19 dose one among female HCW and significantly shorter time to COVID-19 dose one and booster among medical and nursing HCW. This study quantified the extent of booster hesitancy by measuring this by the duration of delay in booster uptake. Booster uptake was found to be temporally unrelated to the local COVID-19 infection rate. Future studies could examine age as a potential factor, and further qualitative studies could help to explore the underlying factors intertwined but correlated with vaccine hesitancy among HCW. Tailored education, improving awareness, risk messaging, and strategic legislation might also help to reduce untimely booster vaccination, as showcased by the different studies conducted worldwide.

## Figures and Tables

**Figure 1 vaccines-10-00464-f001:**
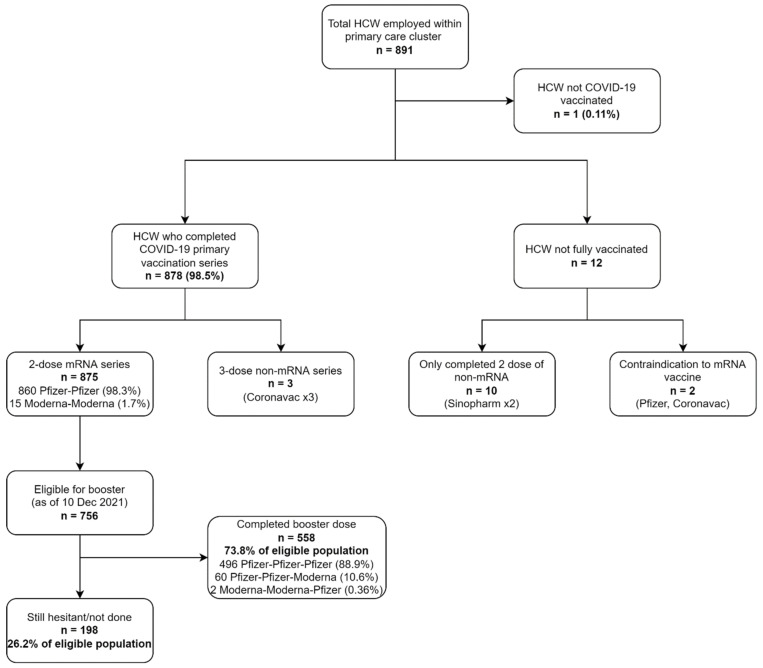
The state of COVID-19 vaccination among HCW.

**Figure 2 vaccines-10-00464-f002:**
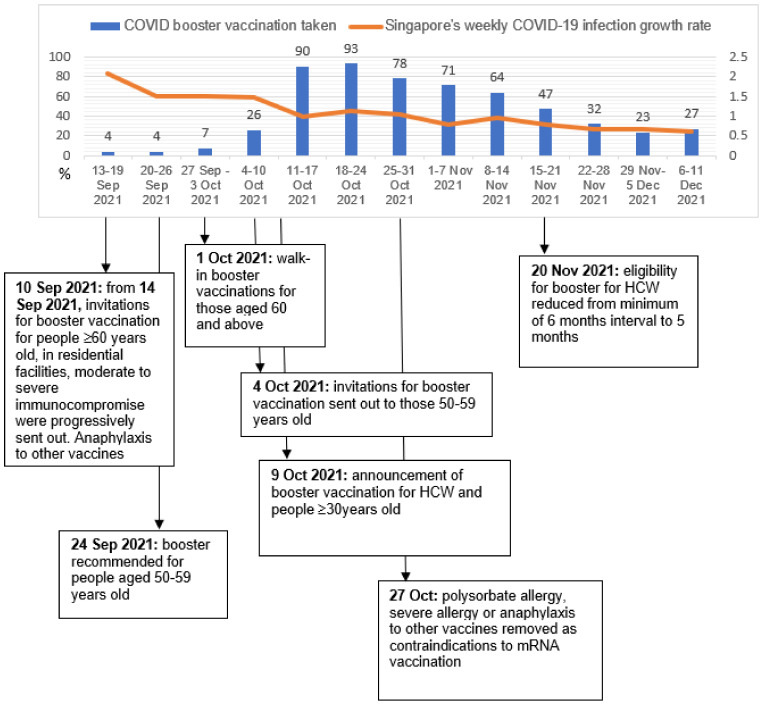
Timeline of COVID-19 booster uptake in relation to policy changes and local infection rates.

**Table 1 vaccines-10-00464-t001:** Mean and median of delay in COVID-19 1st dose and booster uptake among HCW.

Study Variables	COVID-19 Vaccine 1st Dose Delay (Days)				COVID-19 Vaccine Booster Dose Delay (Days)			
	n (%)	Mean (CI)	Median (CI)	*p*	n (%)	Mean (CI)	Median (CI)	*p*
	890 (100%)	69.1 (64.4, 73.8)	39 (32, 46)		558 (62.1%)	28.8 (27.2, 30.4)	24 (21.3, 26.7)	
*Sex*				<0.001				0.782
Female	757 (85.1%)	72.4 (67.2, 77.6)	41 (34.5, 47.5)		466 (83.5%)	29 (27.2, 30.7)	26 (23.2, 28.8)	
Male	133 (14.9%)	50.1 (39.8, 60.4)	19 (12.6, 25.4)		92 (16.5%)	27.4 (23.4, 31.5)	19 (13.3, 24.7)	
*Clinic*				0.072				0.044
Central Office	138 (15.5%)	73.4 (61.8, 85.1)	45 (38.9, 51.1)		87 (15.6%)	31.8 (27.7, 36.0)	30 (23.1, 36.9)	
Clinic A	120 (13.5%)	59.6 (46.2, 73.1)	20 (10.7, 29.3)		76 (13.6%)	29.3 (24.9, 33.7)	24 (12.3, 35.7)	
Clinic B	91 (10.2%)	66.2 (51.4, 81)	25 (14.9, 35.1)		62 (11.1%)	24.7 (20.8, 28.5)	22 (18.2, 25.8)	
Clinic C	128 (14.4%)	62.6 (50.8, 74.5)	20 (8.17, 31.8)		80 (14.3%)	30.9 (26.8, 35.0)	27 (19.4, 34.6)	
Clinic D	88 (9.9%)	72.4 (58.8, 86)	49 (37.5, 60.5)		53 (9.5%)	31.1 (25.6, 36.6)	30 (18.5, 41.5)	
Clinic E	148 (16.6%)	77.7 (65.5, 89.9)	39 (11, 67)		94 (16.8%)	24.9 (21.1, 28.7)	20 (16.1, 23.9)	
Clinic F	104 (11.7%)	55.3 (43.6, 66.9)	31 (13.5, 48.5)		74 (13.3%)	25.9 (21.3, 30.5)	18 (10.3, 25.7)	
Clinic G	73 (8.2%)	89.6 (69.8, 109)	57 (39.3, 74.7)		32 (5.73%)	31.8 (25.1, 38.4)	19 (1.85, 36.2)	
*Profession*				0.019				0.028
Administration *	122 (13.7%)	74.1 (61.2, 86.9)	45 (39, 51)		71 (12.7%)	34.6 (30.3, 38.9)	34 (24.2, 43.8)	
Allied Health ^+^	42 (4.7%)	73.3 (48.9, 97.8)	28 (5.77, 50.2)		21 (3.76%)	34.2 (26.6, 41.8)	33 (20.5, 45.5)	
Ancillary ^	331 (37.2%)	73.9 (66.4, 81.4)	47 (35.6, 58.4)		203 (36.3%)	29.4 (26.7, 32)	26 (22.2, 29.8)	
Medical ^#^	165 (18.5%)	53.3 (43.9, 62.8)	19 (13.4, 24.6)		114 (20.4%)	24.7 (21, 28.4)	16 (11, 21)	
Nursing	230 (25.8%)	70.1 (59.9, 80.3)	22 (12.6, 31.4)		149 (26.7%)	26.9 (23.9, 29.9)	22 (18.8, 25.2)	

* Category includes operation executives and IT support; ^+^ Category includes care coordinators, dieticians, medical social workers, physiotherapists, podiatrists, psychologists; ^ Category includes call centre operators, financial counsellors, patient care and service associates. ^#^ Category includes doctors and dentists.

**Table 2 vaccines-10-00464-t002:** Log rank test for Time to 1st dose among professions.

	Administration		Allied Health		Ancillary		Medical		Nursing	
Profession	Chi-Square	*p*	Chi-Square	*p*	Chi-Square	*p*	Chi-Square	*p*	Chi-Square	*p*
Administration			0.094	0.759	0.022	0.881	7.31	0.007	0.089	0.765
Allied Health	0.094	0.759			0.355	0.551	2.99	0.084	0.280	0.597
Ancillary	0.022	0.881	0.355	0.551			11.7	<0.001	0.005	0.946
Medical	7.31	0.007	2.99	0.084	11.7	<0.001			5.36	0.021
Nursing	0.089	0.765	0.28	0.597	0.005	0.946	5.36	0.021		

**Table 3 vaccines-10-00464-t003:** Log rank test for Time to booster among professions.

	Administration		Allied Health		Ancillary		Medical		Nursing	
Profession	Chi-Square	*p*	Chi-Square	*p*	Chi-Square	*p*	Chi-Square	*p*	Chi-Square	*p*
Administration			0.109	0.742	2.53	0.111	6.17	0.013	6.81	0.009
Allied Health	0.109	0.742			1.87	0.172	3.88	0.049	3.95	0.047
Ancillary	2.53	0.111	1.87	0.172			2.01	0.156	1.79	0.181
Medical	6.17	0.013	3.88	0.049	2.01	0.156			0.116	0.734
Nursing	6.81	0.009	3.95	0.047	1.79	0.181	0.116	0.734		

**Table 4 vaccines-10-00464-t004:** Cox regression for time to COVID-19 booster among HCW.

Study Variables	*B*	*p*	HR	95% CI for HR	
	Coefficient			Lower	Upper
*Sex*					
Female			1		
Male	−0.007	0.958	0.993	0.762	1.30
*Clinic*					
Clinic A			1		
Central Office	0.367	0.08	1.44	0.957	2.18
Clinic B	0.278	0.108	1.32	0.941	1.85
Clinic C	−0.084	0.603	0.92	0.67	1.26
Clinic D	−0.06	0.74	0.942	0.661	1.34
Clinic E	0.258	0.098	1.29	0.954	1.76
Clinic F	0.193	0.238	1.21	0.88	1.67
Clinic G	−0.273	0.197	0.761	0.502	1.15
Profession					
Administration			1		
Allied Health	0.063	0.831	1.07	0.598	1.90
Ancillary	0.427	0.035	1.53	1.03	2.28
Medical	0.588	0.006	1.8	1.18	2.74
Nursing	0.586	0.006	1.8	1.18	2.73

Time to dose 1	−0.003	0.003	0.997	0.994	0.999


HR = Hazard Ratio, CI = Confidence Interval.

## Data Availability

The data presented in the study are available on request from the corresponding author.

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
