# Peer review of "COVID-19 Vaccine Booster Hesitancy among Healthcare Workers: A Retrospective Observational Study in Singapore"

_vaccines, 2022, doi:10.3390/vaccines10030464_

Round 1
Reviewer 1 Report
I have the following comments:
- Line 2 -true should be removed from the title.
- Line 13 – less hesitant than whom?
- Line 36 – this is confusing. How does one go backwards from booster to 1st dose? Can you reword it from 1st dose to booster?
- I wanted to know early who was a HCW. It is defined line 91. Is there a standardized definition? If so, please put definition in introduction. It seems like there should be more information about hesitancy in the literature. Did you look at other types of immunizations?
- In the methods, please use terms inclusion criteria and exclusion criteria.
- Please divide Methods into more than the one subheading. The definitions for (triangle 1 etc.) are line 133 etc. They are not used until 164 and together so a challenge to follow. I recommend words be used for each on 164 etc.
- Line 139 (whichever was later) is confusing.
- Figure 1, first box has total HCW, but that is not the total as you excluded some individuals. This first box is only persons who met inclusion criteria. What would the number be if it included excluded individuals?
- Line 168, please add subject number.
- To me, it is a breach of confidentiality to publish names of clinics (Tables 2 and 3). I highly encourage the removal of clinic names. Can they be joined by maybe type – i.e., private clinic, public clinic, outpatient, inpatient?
- Figure 4 needs descriptors of meaning. I am guessing 0 – 100 are percentages. Possibly the number above each bar is percentage??? What are the values on the right y axis? I do not see 2 bar colors.
- In the Discussion, I encourage Limitations subheading.
- Lines 248-252 – what table or figure are those numbers in?
- References are ok.
Reviewer 2 Report
Dear author(s),
Thank you for your efforts in evaluating the COVID-19 vaccination uptake by HCWs in Singapore which is a very important indicator for public acceptance of vaccines. Nevertheless, there is a number of methodological issues, you need to address.
Major Issues
- The Introduction section fails to provide the rationale for this study methodology.
- What do you mean by "True" hesitancy? What is your reference for this terminology?
- Following your assumption that the patency period between eligibility and actual uptake is the outcome variable that defines "true" hesitancy, how did you control for the too many confounding variables related to medical anamnesis, registration system, vaccination centres capacity, and vaccines availability?
- In the Methods section, please remove the word "Retrospective cross-sectional study". Cross-sectional studies can't be prospective or retrospective. They can either be analytical or descriptive.
Please read this article: https://pubmed.ncbi.nlm.nih.gov/29453895/ - If you consider your study as a cross-sectional study, then you should follow the STROBE guidelines for cross-sectional studies. Please cite the guidelines also in your Methods section.
https://pubmed.ncbi.nlm.nih.gov/17938396/ - In the Discussion section, it would be beneficial to reflect on the potential vaccine hesitancy drivers among healthcare workers and students in other countries, e.g. vaccine effectiveness and side effects.
Suggested refs:
https://pubmed.ncbi.nlm.nih.gov/34072500/
https://pubmed.ncbi.nlm.nih.gov/34301398/
https://pubmed.ncbi.nlm.nih.gov/34696266/
https://pubmed.ncbi.nlm.nih.gov/33367857/
https://pubmed.ncbi.nlm.nih.gov/34579190/
Minor Issues
- Title: the term "True" can be confusing and unnecessary. Please consider changing the title.
- The reference style does not follow the journal guidelines.
- Line 92, remove the capital letters. "Medical and Dental".
- Line 182 - 141, please transform this part into a Table. It will be easier to follow and understand your methods.
Sincerely,
Reviewer 3 Report
Thank you for asking me to review this article. The ongoing pandemic has resulted in global health, economic and social crises. Actually, the vaccination campaign is the first method to counteract the COVID-19 pandemic; however, sufficient vaccination coverage is conditioned by the people’s acceptance of these vaccines in the general population and health care workers. In this context, the paper under review is aimed to identify factors affecting true booster hesitancy by examining actual vaccine uptake across time.
The subject under study is certainly important, especially in the historical period we are experiencing. The article presents interesting results but the manuscript must be improved. I would like to encourage authors to consider several issues to be improved.
Introduction: The authors should make clearer what is the gap in the literature that is filled with this study. The authors do not frame their study within the vast body of literature that addressed the issue of vaccine acceptance and vaccine hesitancy during the pandemic (also if by a questionnaire). What is the national and international situation regarding the acceptance of the vaccination in the adult population?
Methods: The survey was conducted acquiring information retrospectively by consulting a database. The use of an unreliable instrument is a serious and irreversible limitation of the study to ensure face validity, reliability and intelligibility (to be discussed in the limits section). The enrolment procedure must be better specified. How did the authors choose the way to select the sample? All HCW in the database? Why 7 hospitals? What is the total reference population of HCWs? what is the minimum sample size?
Statistical analysis: I suggest to insert a measure of the magnitude of the effect for the comparisons. Please consider to include effect sizes.
Ethical Issue: although anonymously, the authors use sensible and personal data. Therefore, it should be specified if the NHG Domain Specific Review Board is an ethical competent body.
Discussion: I also suggest expanding, emphasizing what is the possible international contribution of the study to the literature. What are the implications of the study? The discussion must be updated including the debated argument of a green pass linked to vaccination practice, if this issue was not considered by the author a paragraph should be added in the limit section with a proper reference (refer to articles with DOI: https://doi.org/10.3390/vaccines9111222).
Round 2
Reviewer 2 Report
Dear author(s),
Thank you for your efforts in improving the presentation of your manuscript.
There is only one point which I can not fully agree about which is the study design. Your study can not be described as a "retrospective cohort" study simply because cohort studies are analytical epidemiologic studies that require comparison between different groups of patients/individuals based on their exposure.
To say that your study was a cohort study, you should be able to define what is the exposure? what is the outcome? and how the subjects of your study were classified according to the exposure (e.g. exposed vs. non-exposed)?
Your study is a case series because your study is descriptive not analytical.
Sincerely,
Author Response
Dear Reviewer, thank you for your input and comments about the terminology of our methodology. We have discussed with our co-authors and agreed that it does not fulfill criteria for a cohort study (lack of control, and the points you have mentioned). After reviewing similar publications and their methodologies (such as reference 23), we decided to term it a retrospective observational study. Thank you.
Reviewer 3 Report
the paper was improved and it is now suitable for publication
Author Response
Dear Reviewer, we thank you for your feedback and kind reviews in aiding this publication.